# Coffee Silverskin as a Functional Ingredient in Vegan Biscuits: Physicochemical and Sensory Properties and In Vitro Bioaccessibility of Bioactive Compounds

**DOI:** 10.3390/foods11050717

**Published:** 2022-02-28

**Authors:** Carolina Cantele, Martina Tedesco, Daniela Ghirardello, Giuseppe Zeppa, Marta Bertolino

**Affiliations:** Department of Agricultural, Forest and Food Sciences (DISAFA), University of Turin, 10095 Grugliasco, Italy; carolina.cantele@unito.it (C.C.); martina.tedesco@unito.it (M.T.); daniela.ghirardello@unito.it (D.G.); giuseppe.zeppa@unito.it (G.Z.)

**Keywords:** coffee silverskin, decaffeination, by-products, polyphenols, in vitro digestion, bioaccessibility, antiradical activity

## Abstract

Coffee silverskin (CS), a by-product obtained by the coffee industry after the roasting process, is scientifically known to be a source of fiber and polyphenols, which could contribute to human health. In this work, the production of CS-enriched biscuits is proposed, where the CS from Arabica and Robusta type and a decaffeinated blend of the two were used at three different levels as a replacement for wheat flour. The biscuits were analyzed for their physicochemical properties, consumer acceptability, and the bioaccessibility of polyphenols after in vitro digestion was estimated in order to identify the formulation most appreciated by consumers and most promising in terms of nutritional and biofunctional potential. From the results, CS-based biscuits represent an interesting possibility to create a more sustainable coffee chain, thanks to the valorization of the silverskin, especially if a decaffeinated CS is considered. In fact, a 4% replacement of the wheat flour with decaffeinated CS is able to give a final product with a high content of accessible polyphenols and a biscuit appreciated by the consumer.

## 1. Introduction

Coffee is one of the most consumed drinks worldwide, having registered about 9 million tons of green coffee beans exported worldwide [1]. The green coffee beans are derived mainly from *Coffea arabica* L., generally known as Arabica, and *Coffea canephora* Pierre ex Froehner, also known as Robusta [1], distinguished by different botanical and chemical characteristics. The fruit of both species is a drupe and, in order to isolate the seeds, industrial processing removes all the other components, which represent more than 50% of the fruit [2]. For this reason, the coffee industry generates an enormous amount of waste and by-products that represent not only a substantial disposal cost for companies, but also an environmental problem. These by-products actually have important bioactive constituents within their composition [1,2,3], which explains the focus of research in recent years on their recovery and valorization.

In particular, coffee silverskin (CS) represents the last component of the drupe to be removed, being detached from the seeds during the roasting process [4], and is characterized by a nutritional profile of particular interest. Costa et al. [5] detailed the chemical composition of CS, including 56.4% dietary fiber (49.1% of insoluble dietary fiber and 7.30% soluble dietary fiber), 18.8% proteins, 8.34% ash including macrominerals for instance K, Mg, Ca, Fe, Na (about 7.61%), 5.80% carbohydrates, 4.76% moisture, 2.42% fat, 1.25% caffeine, 0.25% chlorogenic acids, and 0.004% vitamin E. The primary component of CS, dietary fiber, is mainly composed of cellulose, hemicelluloses, and lignin [4,5,6,7], and the beneficial effects of both insoluble and soluble dietary fiber on human health are well established. In fact, insoluble dietary fiber has been demonstrated to prevent intestinal cancer by improving the intestinal motility [3] and proved in vitro to effectively prevent and treat diabetes by controlling the carbohydrates absorption time [8]. On the other hand, it has been shown that soluble dietary fiber plays a role in decreasing sterol and glucose absorption at intestinal level, reducing serum cholesterol and postprandial blood glucose, and in boosting the calcium absorption at colon and rectum level, also showing a prebiotic activity through the promotion of beneficial gut bacteria [3]. In addition to dietary fiber, CS is rich in chlorogenic acids (CGAs). Among the 30 different species of CGAs identified in coffee beans, the three principal classes are caffeoylquinic acids (CQAs), di-caffeoylquinic acids (diCQAs), and feruloylquinic acids (FQAs) [9]. In particular, CS is distinguished by high content of 3-CQA, 4-CQA, 5-CQA, 3,4-diCQA, 3,5-diCQA, 4,5-diCQA, FQA, 4-FQA, and 5-FQA [2,3,4,5,6,8,9,10]. CS from Robusta has been reported having a greater amount of CGAs than CS from Arabica (68.52 mg/g vs. 11.18 mg/g, respectively) [11]. Chlorogenic acids have been associated with antioxidant, antibacterial, antiviral and anti-inflammatory activities, hepatoprotective and choleretic properties, the ability to modulate the gene expression of many antioxidant enzymes, as well as the ability to inhibit others involved in degradative pathways, with a consequent great impact on the prevention and reduction of diseases, for instance, cancer and cardiovascular and/or neurodegenerative diseases [12]. Other important compounds characterizing CS are melanoidins. Melanoidins are brown color polymers, formed in the Maillard reactions during the thermal processing of foods, and therefore, in the case of coffee, during the roasting of green beans [13]. Since many compounds participate in their formation (i.e., galactomannan- and arabinogalactan-like carbohydrates, polysaccharides, phenolic compounds, and proteins), their chemical structure has not yet been fully clarified [13]. The tendency of phenolic compounds, and in particular chlorogenic acids, to bind with dietary fiber to form a fiber-antioxidant complex (i.e., melanoidins) has been repeatedly highlighted in CS [3,13,14]. Although they were initially considered as anti-nutritional compounds, today numerous beneficial properties are attributed to coffee melanoidins, having been proposed to have antioxidant, antihypertensive, anti-inflammatory, antiglycative, anticarcinogenic, anticariogenic antimicrobial, and prebiotic effects [15]. Hence, as a result of its high content in bioactive compounds, especially polyphenols, CS has been widely claimed as a functional ingredient that can potentially exert beneficial effects on human health by protecting against oxidative damages, carbonyl stress, and accumulation of advanced glycation end-products (AGEs), also possessing a prebiotic activity [3,11,14,16]. However, some of these suggested activities do not always translate into actual biological effects, as their ability to perform some beneficial actions on the human body depends on their ability to reach target cells, tissues, and organs through the circulation system once digested. Successful reaching of the targets is a function of the ability of the compounds to reach the gastrointestinal tract undamaged and released from the food matrix (bioaccessibility) and cross the intestinal barrier to enter the bloodstream (bioavailability) [17].

Besides, CS also contains 0.8–1% of caffeine, which can, on the one hand, help to prevent Parkinson’s and Alzheimer’s diseases by stimulating the nervous system, but, on the other hand, can cause the increase in blood pressure, insomnia, and tachycardia in sensitive individuals [18]. Therefore, in order to reach this target of individuals, the coffee industry can subject the green coffee beans (normally Arabica and Robusta blend), before the roasting process, to a decaffeination procedure to reduce caffeine to a level below 0.3% [19]. The decaffeination could be done using organic solvents such as methylene chloride or ethyl acetate or “natural” solvents such as water or supercritical carbon dioxide (ScCO_2_). Using organic solvents, the stripping agent could be directly in contact with the steamed beans (50% humidity content), or they can be firstly soaked in hot water to dissolve caffeine; afterwards, the water extract is subjected to a decaffeination process in a separate vessel, using the solvent, in order to recycle the water back (indirect way). Using water as a natural solvent, the beans are treated as in the indirect solvent process, but the caffeine is removed from the water extract using a carbon filter. Using ScCO_2_, as in the direct solvent extraction, the moisture of beans is brought to about 50% and then the CO_2_ saturated with water is allowed to recirculate between the beans mass and a scrubber, where the caffeine is condensed [20]. The latter method is the most appreciated, since it allows to separate only the caffeine, minimising the loss of aroma precursor and chlorogenic acids [20].

Thanks to its composition, CS is considered to be a safe nutraceutical ingredient, in the form of both an aqueous extract rich in phytochemicals and an antioxidant fiber [21]; it has already found numerous applications in food products. Martinez-Saez et al. [22] obtained an innovative beverage from Arabica and Robusta CS for body fat reduction and body weight control with satisfactory sensory properties. The aqueous extract of Robusta CS has been tested in the formulation of gluten-free bread, with antioxidant and α-glucosidase inhibition capacity [23]. Grounded CS (Arabica and/or Robusta) has been evaluated in different bakery [24,25,26,27] and dairy products [28], obtaining promising results in terms of nutraceutical value and sensorial characteristics.

Although there is existing research on the use of Arabica and Robusta CS in the formulation of biscuits, there is still, to the best of our knowledge, no work on CS obtained after decaffeination of coffee beans.

The present study thus aims to evaluate the potential of a coffee by-product, the CS, both with and without caffeine, to be integrated in vegan biscuit formulations in terms of technological, sensory, and functional properties.

## 2. Materials and Methods

### 2.1. Chemicals

Methanol (≥99.9%), Folin & Ciocalteu’s phenol reagent, 2-2′-diphenyl-1-picrylhydrazyl (DPPH), 6-hydroxy-2,5,7,8-tetramethylchroman-2-carboxylic acid (Trolox) (97%), α-amylase, pepsin, bile salts, and pancreatin were supplied by Sigma-Aldrich (Milan, Italy). Potassium phosphate monobasic and dibasic, potassium chloride, sodium carbonate and bicarbonate, sodium chloride, magnesium chloride hexahydrate, ammonium carbonate, hydrochloric acid, and calcium chloride dihydrate, all of high purity, were provided by Carlo Erba (Milan, Italy). Caffeine (≥99%), gallic acid (≥98%), ethanol (≥99.9%), and sodium hydroxide (1M) were provided by Fluka (Milan, Italy).

### 2.2. Biscuits Ingredients

Three different types of CS, kindly provided by Lavazza S.p.A., were considered in this study, one obtained from *Coffea arabica* L. (Arabica), one from *Coffea canephora* Pierre ex Froehner (Robusta), and one obtained by mixing and decaffeinating with ScCO_2_ both beans in the proportion normally used by the company. Their nutritional composition determined as per Rojo-Poveda et al. [29] and caffeine content measured with the method reported by Barbosa-Pereira [30] are reported in Table 1.

The particle size of the CSs was reduced to less than 80 µm with Ultracentrifugal Mill ZM 200 (Retsch GmbH, Haan, Germany).

Baking powder, margarine (palm and soybean vegetable oils, water, salt, emulsifier (mono and diglycerides of fatty acids), preservative (potassium sorbate), acidifier (citric acid), coloring agents (carotenes), flavor), oat milk (water, oat (12%), sunflower oil, salt), salt, sucrose, and wheat flour (type 00, corresponding to an ordinary bread-making wheat) were purchased from a local market (Turin, Italy).

### 2.3. Biscuits Formulation and Preparation

The vegan biscuits were produced by replacing wheat flour with each type of CS by 2%, 4%, and 6%. A control was also made, without the addition of silverskin (0%). The ten types of biscuits were named as follows: 0CS (0% of CS, Control); 2CSA, 4CSA, and 6CSA (2%, 4%, and 6% of CS Arabica, respectively); 2CSR, 4CSR, and 6CSR (2%, 4%, and 6% of CS Robusta, respectively); 2CSD, 4CSD, and 6CSD (2%, 4%, and 6% of CS decaffeinated mixture, respectively). Table 2 shows the quantities of the ingredients used for the formulation of the biscuits.

A KitchenAid^®^ dough beater (Whirpool Corporation, Benton Harbor, MI, USA) was used to mix the ingredients. The creaming was produced by whisking the margarine for one minute and then adding the sucrose, salt, baking powder, and oat milk and miking for another two minutes. Lastly, wheat flour and CSs powder were added to the cream and mixed for 3 min. The dough was then rolled out to a thickness of 7 mm and cut into 6 cm diameter circular shapes and finally baked in an OLIS oven 044-054S (Ali Group, Milan, Italy) in fan mode for 15 min at 180 °C.

The biscuits were ground with a Ultracentrifugal Mill ZM 200 (Retsch GmbH, Haan, Germany) to reduce their particle size to less than 1 mm.

### 2.4. Physicochemical Analyses

Moisture was measured on 5 g of biscuit powder, using a MAC210/NH thermo-moisture analyzer (Radwag, Radom, Poland) at 130 °C. Water activity was measured on 2 g of biscuit powder using the AcquaLab PRE water activity meter (Decagon devices, Pullman, WA, USA). Color was evaluated through the CIELAB color space indices L*, a*, and b* on biscuit powders, using a CM-5 spectrocolorimeter (Konica Minolta, Tokyo, Japan) on transmittance with a measurement area of 8 mm, an angle of observation of 10 °, an illuminant D65, a wavelength spectrum between 360 and 740 nm, and in SCE (Specular Component Excluded) mode. The indices were used in order to calculate the hue (*h_ab_*), expressed as degrees (°), and chroma (*C*_ab_*) using the following equations:*h_ab_* = arctan (b*/a*)
*C*_ab_* = [(a*)^2^ + (b*)^2^]^1/2^

Moreover, the CIELAB values were used to evaluate the ΔE parameter, as indicated by Rojo-Poveda et al. [31], which enables to evaluate whether (ΔE > 2.5) or not (ΔE < 2.5) the human eye is able to distinguish between two samples on the basis of colour. 

### 2.5. Structural Analyses

Weight loss, spread ratio, and hardness were performed as described by Rojo-Poveda et al. [31].

Weight loss (g) was calculated weighting twelve biscuits before and after baking. The spread ratio was measured using 6 biscuits per type applying the AACC 10-50.05 method [31]. The hardness was measured using a 25-kg cell loaded TAXT2i Texture Analyzer^®^ (Stable Micro Systems, Godalming, UK), and a HDP-BS cutting blade, applying a compression test speed of 1 mm/s. The results, expressed as the maximum force (N) necessary to break the biscuit, were acquired using a Texture Expert Exceed software package for Windows (version 2.54) (Stable Micro Systems, UK).

### 2.6. Consumer Acceptance Evaluation

A consumer acceptance test was made on the biscuits using a nine-point hedonic scale form, where 1 indicated maximum dislike, 5 correspond to neither like nor dislike, and 9 indicated maximum appreciation [32]. Forty-eight untrained participants (gender and aged balanced between 21 and 65 years old) were enrolled and required to evaluate the biscuits in terms of appearance, odor, taste, flavor, texture, overall liking, and purchase predisposition (the latter rated from 1 to 7, where 1 corresponded to the minimum, 4 correspond to neither willing nor not, and 7 to the maximum purchase predisposition—Appendix A). The samples were randomly and evenly presented to the tasters, with still water provided to rinse their mouths between them. After collecting written informed consent, the test was conducted in a room at 21 °C equipped with white lights.

### 2.7. Determination of Total Phenolic Content

The total phenolic content (TPC) of the biscuits was assessed before and after in vitro GID by the Folin & Ciocalteu’s colorimetric method adjusted to a BioTek Synergy HT spectrophotometric microplate reader (BioTek Instruments, Milan, Italy) [30]. Each sample was analyzed in triplicate. Calibration curve (100–500 µM) was built for quantification using gallic acid as standard. Results were expressed as mg of gallic acid equivalents (GAE) per gram of biscuit.

### 2.8. Determination of Radical Scavenging Activity

The radical scavenging activity (RSA) was assessed before and after in vitro GID through the DPPH free radical inhibition method detailed in Barbosa-Pereira et al. [30] by using a BioTek Synergy HT microplate reader. A wavelength of 515 nm was selected to measure the decrease in DPPH free radical absorbance. Each sample was analyzed in triplicate. Quantification was achieved by building a calibration curve of standard Trolox (12.5–300 µM) and the results were reported in μmol of Trolox equivalents (TE) per gram of biscuit.

### 2.9. In Vitro Simulated Gastrointestinal Digestion

The ten types of biscuits were subjected to in vitro simulated gastrointestinal digestion (GID), as detailed in the standardized static method proposed by Minekus et al. [33]. The digestive protocol comprised of three phases (oral, gastric, and intestinal), each with its own corresponding simulated digestive fluid (simulated salivary, gastric, and intestinal fluids, respectively), consisting of electrolytes, enzymes, and water. At first, 5 g of biscuits were minced to simulate the mastication. Then, the digestive fluids were added to the sample and left stirring continuously at 37 °C for 2 min for the oral phase and for 120 min for both gastric and intestinal phases. At the end of the intestinal phase, the digestive process was ended lowering the pH to 5.4, then the samples were immediately centrifuged (12,500× *g*, 4 °C, 10 min) and the supernatant was filtered at 0.45 μm with cellulose acetate filters. Until further analysis, the samples were preserved at −20 °C.

The in vitro bioaccessibility (%) of bioactive compounds was calculated as follows:Bioaccessibility = (Conc_post_/(Conc_pre_) × 100
where Conc_post_ and Conc_pre_ represent the TPC (mgGAE/g biscuit) at the end and at the beginning of the process, respectively.

For undigested samples, in order to extract the phenolic compounds, the same procedure was applied, replacing the simulated digestive fluids with pure water.

### 2.10. Statistical Analysis

Data obtained from the physicochemical analyses, TPC, and RSA were analysed by one-way ANOVA with Duncan’s post hoc test at a 95% confidence level using SPSS Statistics 26 software (IBM-SPSS, Inc., Chicago, IL, USA).

The Kruskal–Wallis test (test H) and one-way ANOVA were used to process the values obtained by the consumer acceptance test using the SPSS Statistics 26 software (IBM-SPSS Inc., Chicago, IL, USA).

## 3. Results and Discussion

### 3.1. Physicochemical Characterization

Moisture, water activity (a_w_), CIELAB values (*L**, *h*_ab_, *C**_ab_), and ANOVA results are reported in Table 3. Significant differences (*p* < 0.001) were found between the control and the biscuits added with CS for all the parameters under evaluation.

Regarding moisture, the control displayed a value of 4.27 ± 0.18, while the CS-added biscuits reported percentages between 5.44 ± 0.12 and 7.38 ± 0.15. In detail, a linear increase in CSA biscuits was observed as the percentage of CS—included in the formulation—increased with a value of 3% between 2CSA and 4CSA and 8% between 4CSA and 6CSA, while in CSD biscuits, an opposite behaviour was found, with a decrease—a value of 5% between 2CSD and 4CSD and 9% between 4CSD and 6CSD. On the contrary, CSR biscuits showed higher values of moisture with respect to 2% when added with 4% of CS, but it dropped when 6% of CS was added. Comparing the different types of CS, in general, CSA biscuits were characterized by the lowest values, whereas CSR biscuits had the highest. Similar behaviour was observed for a_w_, which accounted for 0.31 ± 0.00 in the control and between 0.41 ± 0.01 and 0.52 ± 0.00 in the CS-added samples. As a_w_ represents the free water content of a food product and therefore the water available for chemical, physical, and enzymatic reactions, it is desirable that it remains at low values (below 0.60), as well as moisture, which should always remain below 10% in order to avoid microbial spoilage [34,35]. Since all the samples displayed values below the threshold of acceptability, it can be concluded that the addition of the by-product under examination should not lead to a consistent deterioration of the product from the point of view of microbiological safety. The trend found in CSA biscuits agrees with that observed by Gocmen et al. [26], who replaced wheat flour in biscuits with 2.5%, 5%, and 7.5% of CS and Rojo-Poveda et al. [31] who formulated biscuits by replacing wheat flour with cocoa bean shell. It is likely that the high fibre content of CS, especially the soluble fraction, plays a key role, leading to a higher water retention capacity and thus a higher moisture content. On the contrary, Mahloko et al. [35] observed that in biscuits formulated with banana peel flour and prickly pear flour, the moisture content decreased. This was explained by the lower moisture content in banana peel and prickly pear flours, compared to wheat flour [35]. The same situation was found by Agu and Okoli [36] in biscuits with different percentages of combining beniseed (BF) and unripe plantain flours (UPF) replacing wheat flour. As the percentage of beniseed and plantain flours increased, the moisture content decreased progressively, the decrease being particularly evident in the formulation containing the highest amount of BF and UPF [36]. The chemical composition of the CS used in this study revealed that CSA, compared to CSR and CSD, was characterized by a higher water and fibre content that in synergy gave a more humid biscuit, confirming this hypothesis. In CSR and CSD, the opposite situation occurred, probably due to the fact that their composition had lower moisture and fibre values than CSA, which was reflected in the moisture and a_w_ values.

With regard to colour (Figure 1), significant differences (*p* < 0.01) were found for all the CIELAB parameters (*L**, *h_ab_*, and *C*_ab_*) between 0CS and the biscuits added with CSA, CSR, and CSD. 

The hue angle of 0CS was quantified in 70.78 ± 0.44, corresponding to orange on the colour wheel. The addition of silverskin led in all cases to a significant variation of its value. The addition of 2% CSA increased the hue by 1 degree toward yellow, and this is not surprising considering the colour of the by-product itself, which was light coloured and yellowish (Appendix A); the further addition of CS beyond 2% did not result in a further change in biscuit hue. However, for both Robusta and decaffeinated blend, all percentages affected this parameter. In particular, the addition of 2% CSR increased the angle by 2 degrees, bringing the hue towards the yellow colour, and then gradually decreased back towards the value observed for the control. In CSD, on the other hand, the change was more pronounced than in the other two types, with a gradual decrease from 0% CS to 6% CS of almost 6 degrees, thus tending towards red-orange. The lightness of the biscuits was clearly affected by the presence of all the type of CS, with a significant decrease as the percentage of CS was incremented. Arabica was the type of CS that had the lowest impact on this parameter, causing a decrease of 17% from 0% to 6% of CSA. Meanwhile, compared to the plain biscuit, the addition of 6% CSR led to a decrease in *L** value of 33%, almost twice as much as CSA. Nonetheless, the most massive effect can be attributed, by far, to 6CSD, which saw its brightness drop by almost half, compared to 0CS (decrease of 46%). More than due to Maillard browning, this phenomenon can be traced back to the nature of the by-product itself, as CSA was naturally characterized by a light coloration, CSR by a darker coloration but lighter than CSD, which was indeed dark coloured (Appendix A). Interestingly, the decrease of *L** within the percentages of each CS was found to be steady for CSA and CSR. In fact, with each progressive addition of 2% of CSA (from 0% to 2%, from 2% to 4%, and from 4% to 6%), there was a lowering of *L** amounting to 6%. This also occurred for CSR, where, however, the gradual decrease amounted to 13% for the first two additions and 11% for the last. On the contrary, in CSD, the loss of lightness was not constant but faded as the amount of CS increased (decrease equal to 24% from 0% to 2% CS, 19% from 2% to 4% CS, and 12% from 4% to 6% CS. A similar situation was observed by Ates and Elmaci [27] in cakes prepared with CS at three different rates (20%, 25%, and 30% on the total amount of wheat flour). The values for brightness progressively decreased. These findings were also in agreement with that of Gocmen et al. and with Garcia-Serna et al. [25,26], where *L** decreased with increasing percentage of CS, resulting in darker biscuits. Likewise, chroma also showed a decrease in its value as the percentage of all three silverskin types increased, and again, Arabica experienced the slightest decrease, while the decaffeinated blend’s change was considerable. In fact, compared to the control, the change in chroma amounted to 13% for CSA, 23% for CSR, and 39% for CSD. Thus, it can be said that the addition of silverskin in the biscuit formulation leads to greying of the cookies, making them dirtier.

With regard to the ΔE parameter (Appendix A), the first assessment was made in relation to the control, where all samples showed a ΔE greater than 2.5, ranging 4.19–29.47, with 2CSA displaying the lowest value, while 6CSD had the highest. Therefore, all CS-added biscuits were visually recognizable from the biscuit without CS, even at the lowest integration concentration. Interestingly, when analysing the different percentages, the ΔE was greater in all three CS origins, when comparing 2% and 4% than when comparing 4% and 6%, emphasizing that increasing CS above 4% probably does not further affect product colour in a way that is perceived by consumers. Finally, when comparing the three varieties of CS, the biscuits whose difference in colour was most pronounced were CSA and CSD biscuits, with an increasing in ΔE as the concentration of CS increased. In conclusion, these results suggest that CS could be used as a natural colouring agent in bakery products, as it has a great influence on the characteristic browning of bakery products.

### 3.2. Structural Characterization

Results of weight loss after baking, spread ratio, and hardness, used to describe the structural behaviour of the biscuits after the integration of CS in their formulation, are presented in Table 4.

The weight loss did not provide statistically significant results (*p* > 0.05), thus indicating that the increase of the level of the by-product in the formulation did not influence the amount of water lost as a result of the cooking process. These results agree with Rojo-Poveda et al. [31], where no significant differences were noticed in biscuits formulated with three different percentages of cocoa bean shell (CBS) powder (0%, 10%, and 20%), and two different sugars (sucrose and tagatose). Comparing the different types of CS, the only significant difference was found at rate of 6%, with 6CSA showing the lowest weight loss also compared to 0CS, followed by 6CSR and 6CSD, which exhibited the highest value.

Regarding the spread, which clarifies the behaviour of the biscuit to develop in terms of width or rather in thickness, all the three different CS showed lower values with respect to the control (*p* < 0.01). 0CS was characterized by a spread ratio equal to 6.49, while in the CS-added biscuits, it ranged 4.15–5.43. In both CSA and CSR biscuits, no significant differences (*p* > 0.05) were found between the three percentages of CS. For the decaffeinated blend, on the contrary, the biscuits added with 4% and 6% of CS showed higher spread ratios than 2CSD. The decrement in spread ratio when CS was added in biscuits was not unexpected and can be attributed to the high fibre content of CS. In fact, it has been widely reported that the increase of hydrophilic compounds like fibre leads to a decrease in width and an increment in thickness; the fibre, in fact, compete for the free water present in the dough with the gluten proteins due to their strong water-binding capacity, thus preventing the complete development of the gluten network [26,31]. Our results are in line with those obtained by Gocmen et al. [26] in cookies with CS added at 2.5%, 5%, and 7.5%, and with those obtained by Rojo-Poveda et al. [31] in biscuits with cocoa bean shell added at 10% and 20%, showing a decrease in the spread as the percentage of integration increased. Ojha and Thapa [37] reported a reduction of the overall spread ratio as well with respect to the control when the percentage (3%, 6%, and 9%) of mandarin peel powder increased in biscuits. On the contrary, Mahloko et al. [35] found that, compared to a wheat flour biscuit (control), the spread ratio increased with the addition of banana peel and prickly pear flours in the formulation, and the authors concluded that the latter were less hydrophilic than the starch present in wheat flour. When the three different types of CS were compared, no significant differences were found, except at 2% integration level. In fact, 2CSR revealed the lowest spread ratio, and therefore the greatest development in terms of thickness and compactness, followed by 2CSD and 2CSA, characterized by the highest spread ratio among all the samples.

As for the textural analysis, the maximum force required to break the biscuit, represented by the hardness parameter, was measured. The control (0CS) displayed a value equal to 64.92 ± 8.91 N; on the other hand, when CS was added in the formulation, the hardness of the biscuits ranged between 65.85 N and 104.76 N. Overall, the addition of CS in the biscuit formulation did not substantially affect their hardness. Indeed, in CSR biscuits, no significant differences were shown among the four products (0CS, 2CSR, 4CSR, and 6CSR) (*p* > 0.05), while in CSA and CSD biscuits, only the 4% sample was found to have a higher hardness than the control and the biscuits at 2% and 6% CS (*p* < 0.05). Significant differences (*p* < 0.001) between the three types of CS were only found at 4% concentration, where CSD biscuits were reported to have the highest hardness followed by CSA and finally CSR products. The increase in hardness when CS was added at 4% might be due to the absorption of water by the fibres, which reduces the water available for the development of the gluten network, resulting in an increase in hardness and a loss of crispness [31]. Interestingly, however, when adding the greatest percentage of CS (6%), no such increases in hardness value were detected, as expected, since a higher percentage of CS implies a higher percentage of fibre. Our results disagreed with those found by Rojo-Poveda et al. [31], which in contrast revealed that the higher the cocoa bean shell added in the formulation of biscuits, the higher was the hardness.

As reported by Mahloko et al. [35], the differences that were observed could be due to the structure of the biscuits as a consequence of the addition of the ingredients used, which could lead to fluctuations in maximum force. This could also be due to the mixing phase, which distributes the added ingredients in a way that favours the absorption of water to the disadvantage of an accurate development of the dough structure. The development of the gluten network contributes to the hardness of the biscuits.

### 3.3. Consumer Acceptance Evaluation

Results of the consumer acceptance evaluation are depicted in Table 5. Since this sensory test was carried out with the aim of identifying among all the biscuits tested the one most appreciated by consumers, in this case, statistical analysis was performed among all the samples as a whole, and not taken individually by CS type or integration percentage. Significant differences (*p* < 0.001) were obtained for all parameters evaluated, meaning that the addition of CS led to an overall sensory change of the biscuits. In particular, for appearance, the highest rank sum was calculated for the 0C sample (average score 7.50 ± 0.65—Appendix A), followed by 2CSA (average score 7.50 ± 0.51), and the lowest being 6CSR (average score 6.17 ± 1.64). Taking into consideration the results of the colour analysis, it emerges that the biscuits assessed by the tasters as the best from the point of view of aspect were those characterized by the highest brightness and chroma. Regarding odour, it can be seen that the addition of silverskin did not, in any case, lead to an improvement in olfactory perception. In fact, compared to the control (average score 6.67 ± 1.12), biscuits added with both Arabica and Robusta CS received lower scores at all percentages (average scores among the percentages equal to 6.06 ± 0.05 and 5.67 ± 0.30 for CSA and CSR, respectively), with a more pronounced trend in CSR as the percentage of by-product increased. In contrast, however, the decaffeinated CS blend led to an improvement over 0CS, also achieving the highest score among all samples in the case of the 2CSD (average score 6.83 ± 1.15). This discrepancy in the odour appreciation of the biscuits could be due to the formation of unpleasant volatile compounds during the Maillard reactions in Arabica and Robusta due to their chemical composition that, when used in blend, are mitigated. In fact, the development of both flavours and off-flavours occurs at the expense of nitrogen-containing compounds such as amino acids, peptides, proteins, serotonin, and trigonelline; and hydroxy-acids, reducing carbohydrates and phenols, which are very different between coffee varieties [38]. CSD biscuits obtained the highest scores in terms of taste as well, with 2CSD and 4CSD being the most highly rated (average scores 7.42 ± 0.96 and 7.42 ± 1.05, respectively), which was also reflected in the flavour as well (average score 7.25 ± 1.10 and 7.42 ± 1.20, respectively). As for texture, the addition of CSA led to a worsening of the textural perception of the biscuits, showing lower scores than the control (6.36 ± 0.24 vs. 7.17 ± 1.00 for CSA and control, respectively), while in the case of Robusta, the samples were rated on par with the reference. In contrast, however, CSD also significantly improved this aspect, with 4CSD being the most appreciated by the tasters (average score 7.67 ± 0.86). Hence, the fact that the CSD biscuits showed higher values of hardness did not negatively influence the overall acceptance, but rather improved it. The abovementioned high scores achieved by CSD biscuits consequently translated into high scores with respect to overall liking and purchase predisposition. In fact, 2CSD was the most overall liked (average score 7.33 ± 0.75), immediately followed by 4CSD (average score 7.25 ± 0.93), which was nonetheless the most prone to be purchased by consumers (average score 5.75 ± 0.93). On the other hand, the biscuits added with CSA were the least appreciated by tasters, getting the lowest scores (average score 6.08 ± 0.14), while CSR-added biscuits were deemed on par with the control (average score 7.25 ± 0.93), except for 6CSR in terms of overall liking, which obtained a slightly lower score (average score 6.17 ± 1.59). The low scores obtained for CSA in terms of taste and flavour, which were then reflected in the overall liking, could be due again to the development of unpleasant non-volatiles during Maillard reactions and/or for the natural presence of some compounds in CS. These compounds include caffeine, which contributes to bitterness, polysaccharides that retain volatiles, trigonelline, humic acids, chlorogenic acids, which contribute to astringency, and most of all carboxylic acids such as malic, citric and acetic, which contribute to sourness and are known to be abundant in the Arabica variety [38]. Hence, overall, compared to the control, CSA led to a worsening of sensory characteristics, CSR did not make substantial changes, and CSD improved them.

### 3.4. Total Phenolic Content and Radical Scavenging Activity

The mean values of the total phenolic content (TPC) and the radical scavenging activity (RSA) evaluated on CS-added biscuits are reported in Table 6.

As for the TPC, the three different CSs showed different behaviours, but in all cases, the integration of CS and its increase in the formulation led to a gradual rise in the TPC of the biscuits. In particular, the biscuits added with CS displayed a TPC ranging 0.42–0.49 mg GAE/g, 0.41–0.57 mg GAE/g, and 0.72–1.36 mg GAE/g for CSA, CSR, and CSD biscuits, respectively, with a starting value of the control equal to 0.41 ± 0.02 mg GAE/g. For both CSA and CSR biscuits, significant differences (*p* < 0.001) with respect to the control were only found when adding CS from 4% upwards. The 6CSR was characterized by a higher TPC than CSA (*p* < 0.001). Indeed, CSR biscuits registered a greater increment in TPC from 0% to 6% of CS compared to CSA biscuits (39.0% vs. 19.5%, respectively). On the other hand, the decaffeinated blend showed the highest TPC values at each concentration, with a significant (*p* < 0.001) increase with respect to the control already at 2% of CS, and with sample 6CSD exhibiting three times the initial content of polyphenols. Similar results were obtained in biscuits added with 1% and 3% of green and black tea, ranging 0.27–1.17 mg GAE/g [39]. Thus, CS provided this bakery product with the same polyphenol content as one of the most polyphenol-rich plant materials. Higher results were obtained in biscuits with spent coffee grounds (1.53–2.06 mg GAE/g), which were, though, added at higher concentrations than those used in this study (i.e., 10%, 17%, and 25%) [40]. It should be, however, pointed out the limitation of this analytical method is that it can only give us an idea of the content of polyphenols of a sample, and not a precise measure. Indeed, the overestimation that affects this analysis has been confirmed several times, as the reagent is not selective for phenolic compounds, but it is also able to react with other compounds, such as reducing sugars and organic acids [41].

The RSA increased as the CS concentration in the biscuit increased. In particular, RSA values rose by 323.3%, 65.1%, and 690.7% in 6CSA, 6CSR, and 6CSD, respectively, compared to the control. Nevertheless, while the decaffeinated blend proved again to be the best performing, CSA and CSR biscuits showed a different and opposite trend to that observed for TPC. In fact, values up to four times higher than CSR were found for CSA products. We believe this might be due to the different antiradical activity of the phenolic compounds that were likely to be extracted with water. Indeed, while it is true that polyphenols possess antiradical activity, it is also a fact that this property mostly depends on their structure, and in particular on the number of hydroxyl groups on the aromatic rings, thus leading to a more or less marked activity [42]. Hence, one possibility could be that water extraction only allowed phenolic compounds with a lower antiradical activity to be extracted in CSR biscuits. It is important to recall the fact that extraction with water was carried out in this study to make it possible to study bioaccessibility and thus compare undigested and digested samples. It is in fact well known that water extraction is not the most effective method for bioactive compounds compared to organic solvents [43]. Another possibility could be related to the degradation of chlorogenic acids. In fact, even though the CSR generally contains a greater amount of chlorogenic acids compared to the CSA, in this coffee matrix, they appear to be more sensitive and more easily degradable [44]. Kamiyama and colleagues [45] demonstrated that during roasting, 5-caffeoylquinic acid—one of the most abundant chlorogenic acids in CS—is easily prone to degradation, resulting in the formation of volatile antioxidants. Therefore, the low values of RSA in CSR biscuits might be ascribed to a considerable loss of caffeoylquinic acids during the baking of the biscuits due to the high temperatures, which eventually also removed the volatile antioxidants formed. However, no characterization of polyphenols profile has been conducted in this study, and thus we cannot draw firm conclusions but only hypotheses.

The higher polyphenols content and antiradical activity observed for CSD biscuits may be possibly credited to melanoidins. Melanoidins are formed as a result of Maillard reactions between phenolic compounds and carbohydrates, dietary fibre, and proteins of the coffee beans during the roasting phase [3]. These newly formed compounds abound in phenolic groups, which still provide strong antioxidant activity [3]. In addition, after decaffeination with supercritical carbon dioxide, the green coffee beans take on a darker colour; it has been demonstrated that this brown colour is not the result of the oxidation of the polyphenols, but rather of the formation of melanoidins [46]. Thereby, in CSD, an accumulation of melanoidins may have occurred during both the decaffeination and roasting phases, thus raising the TPC and RSA values compared to CSA and CSR biscuits.

### 3.5. Bioaccessibility of Bioactive Compounds

All the biscuits were in vitro digested in order to assess the bioaccessibility of the bioactive compounds and therefore their actual ability to scavenge free radicals, comparing the values of TPC and RSA obtained before GID (pre-GID) with those obtained after GID (post-GID). Results are depicted in Figure 2 and reported in detail in Appendix A.

Overall, all digested samples showed significantly higher polyphenol content and antiradical activity than their undigested counterparts (increasing of the values from 264% upwards) (*p* < 0.001), with bioaccessibility largely exceeding 100% in all cases, including the control.

Concerning TPC, in 0CS, the post-GID value (1.49 ± 0.08 mg GAE/g) was almost four times that of pre-GID, with a bioaccessibility percentage of 365%. Again, for all three types of CS, the increase in their concentration in the biscuits led to an enhancement of the TPC. As already seen for the undigested samples, the biscuits obtained with Arabica and Robusta CS showed similar values after GID (Figure 2), with a significant increase in the TPC with respect to 0CS only at 4% of CS and above (*p* < 0.001). At 6% CS concentration, post-GID values for CSA (1.81 ± 0.01 mg GAE/g) and CSR (1.88 ± 0.01 mg GAE/g) biscuits were found to be 22% and 26% higher than the control, respectively. Finally, in line with the results obtained before in vitro digestion, the decaffeinated blend was found to have the highest amount of polyphenols, ranging 2.04–3.19 mg GAE/g in the three concentrations of CS. Already at 2% of CS, CSD biscuits showed an increase in TPC compared to the control of 37%, which rose to 68% and 114% at 4% and 6% of CS, respectively.

Gocmen and al. [26] found that the bioaccessibility of phenolic compounds after in vitro digestion of biscuits with 2.5%, 5%, and 7.5% of Arabica CS only ranged from 84% to 89% (56% in the control). On the contrary, similar results to this study were obtained for wheat pasta added with berry fruits and partially-deoiled chia flour and for whole-wheat flour pasta [47,48,49]. The dramatic increase in values following in vitro digestion is attributable to the food matrix and its effect on phenolic compounds. In fact, phenolic compounds are able to interact with other food constituents such as lipids, carbohydrates, and proteins, which can either establish chemical bonds or trap them in their porous structures [50]. Chemical bonding includes non-covalent (hydrophobic interactions, hydrogen bonds and Van der Waals forces) and covalent bonds, weak or strong, reversible or irreversible, depending on the structure of the compounds [50,51]. The amount of bioactive compounds that are accessible, and thus available for absorption, depends primarily on the extent to which they are released from the food matrix, which therefore plays a crucial role in their fate in the human body [52]. Liquid food matrices provide directly accessible bioactive compounds, whereas compounds present in solid matrices must first be released from the latter before they can be absorbed in the gastric or intestinal tract. However, the fact that they are directly accessible means that bioactive compounds in liquid foods are more susceptible to degradation during the digestive process by enzymes, salts, and pH, as was demonstrated in our previous study on beverages made from cocoa bean shell [42]. On the contrary, the link between polyphenols and macro-components of a food, such as carbohydrates, fiber, and proteins, may lead to their protection from degradation pathways [51]. We therefore hypothesize that the low TPC values obtained in the undigested biscuits could be due to the bonds formed between the polyphenols (both in the silverskin and in the flour) and other food constituents, in particular the gluten network and the dietary fiber. Once digested, the pH conditions and the contact with enzymes allowed these complexes to break down, releasing the phenolic substances.

Similar results were found for the RSA. After GID, the control showed the lowest antiradical activity, namely 2.98 ± 0.12 μmol TE/g, which was seven times that of the undigested counterpart, registering a bioaccessibility of 745%. The addition of CS in the formulation of the biscuits led to an improvement of the antiradical activity, with again the exception of samples 2CSA and 2CSR, which showed no significant differences compared to the control, but only a slight tendency to increment (*p* < 0.05). Interestingly, despite CSR biscuits being characterized by the lowest RSA before GID, they displayed similar or even higher values than CSA biscuits after the digestive process. It is likely that the bioactive compounds in the CSR products were more affected by the food matrix but were released later and could therefore exert their antiradical activity. Once again, CSD biscuits exerted the highest RSA, reaching 6% of CS 13.39 ± 0.36 μmol TE/g, and with bioaccessibility ranging 394–625% in the three CS concentrations.

Considering the average weight of a biscuit present in the market (12 g), the doses of phenolic compounds that could be absorbed by the human body after GID of the studied biscuits are promising. The control would provide the lowest quantity, namely 17.87 mg GAE. CSA and CSR biscuits would supply 19.19–21.71 mg GAE and 19.25–22.53 mg GAE, respectively, for the three CS percentages. The highest dose would be delivered by decaffeinated biscuits, with 24.46 mg GAE (2% of CS), 30.03 mg GAE (4% of CS) and 38.23 mg GAE (6% of CS) of bioaccessible polyphenols. However, as already discussed, the analysis of total phenolic content by Folin-Ciocalteu’s reagent is known to be affected by interferents, causing an overestimation of the results [41]. Therefore, it would be necessary to profile the polyphenols contained in the biscuits with chromatographic techniques in order to clarify the real in vitro bioaccessibility. Moreover, in vitro studies are not sufficient to reach solid and reliable conclusions in these terms, but only provide preliminary guidelines. For this reason, further in vivo studies are required in order to determine to what extent these CS-added biscuits are able to provide the human body with concrete benefits.

## 4. Conclusions

The aim of this work was to evaluate the feasibility in technological terms and the potential nutritional benefits of Arabica and Robusta CS when added to the formulation of vegan biscuits. In addition, to the best of our knowledge, CS from the decaffeination of a blend of coffee beans of the two aforementioned coffee was studied for the first time in these terms. CS influenced the chemico–physical characteristics of the biscuits, as well as their polyphenol content and their bioaccessibility and antiradical activity. In fact, being rich in fiber, CS led to an increase in moisture content, water activity and, in some cases, hardness, and to a decrease in the spread ratio, while still remaining within acceptable limits. Moreover, as the percentage of CS within the biscuits increased, the total polyphenol content increased linearly. After in vitro simulated gastrointestinal digestion, the polyphenol content was much higher, with bioaccessibility values largely exceeding 100%. The results of the total polyphenol content were then reflected on the antiradical activity, with values after in vitro digestion also in this case dramatically higher than those obtained on the undigested. In particular, biscuits added with decaffeinated CS proved to be the most promising from a technological, sensory, and nutritional point of view. In fact, they were sensorially satisfactory, being favored over all the other biscuits by tasters, and showed the highest values of total polyphenol content and radical scavenging activity, even after in vitro digestion. This work therefore demonstrates the enormous potential of CS to be used in the formulation of bakery products with possible benefits on human health, especially the decaffeinated blend, which can provide a high content of bioactive compounds, while avoiding the possible negative effects of caffeine. However, other investigations will be necessary to verify these plausible beneficial properties of CS in vivo as well.

## Figures and Tables

**Figure 1 foods-11-00717-f001:**
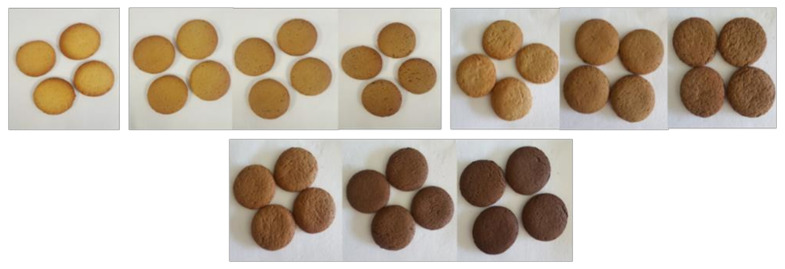
Visual comparison of the color between the control and the biscuits added with 2%, 4%, and 6% of CSA (Arabica coffee silverskin), CSR (Robusta coffee silverskin), and CSD (decaffeinated coffee silverskin). From left to right: 0C; 2CSA, 4CSA, 6CSA; 2CSR, 4CSR, 6CSR; 2CSD, 4CSD, 6CSD.

**Figure 2 foods-11-00717-f002:**
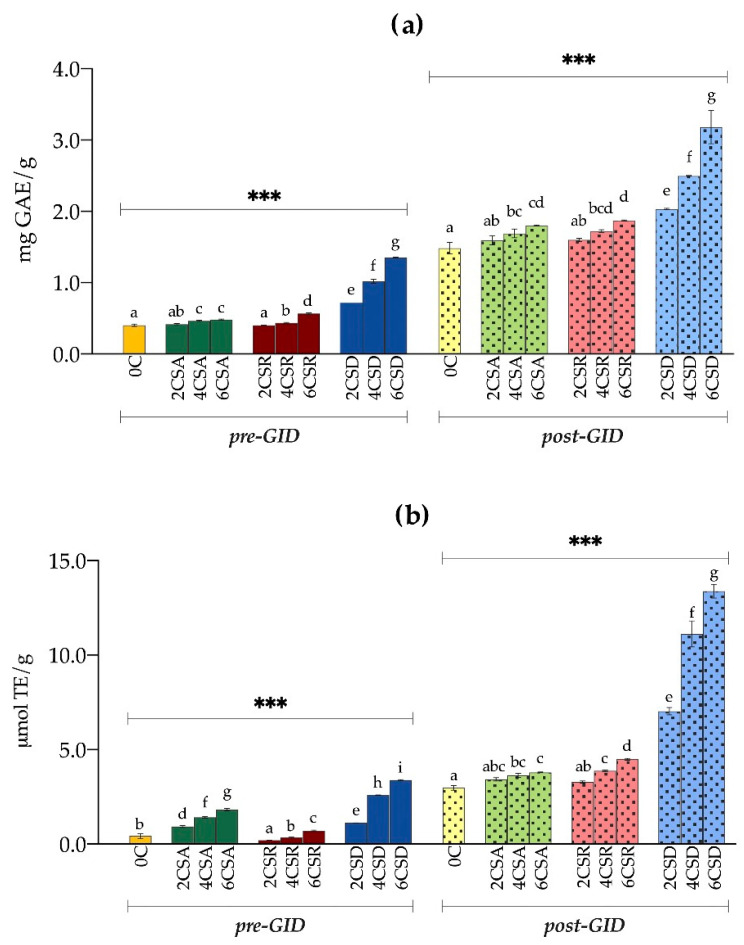
(**a**) TPC (mg GAE/g) and (**b**) RSA (μmol TE/g) of the biscuits before (pre-GID; bars with no pattern) and after (post-GID; bars with dot pattern) gastrointestinal digestion (GID); CSA, Arabica coffee silverskin; CSR, Robusta coffee silverskin; CSD, decaffeinated coffee silverskin. Results of ANOVA with Duncan’s post hoc test are reported within the different types of biscuits before GID and after GID. Bars with different lower-case letters are significantly different at *p* < 0.05. Significance: *** = *p* < 0.001.

**Table 1 foods-11-00717-t001:** Chemical composition of the three different types of coffee silverskin (Arabica, Robusta, and decaffeinated blend).

	Arabica	Robusta	Decaffeinated
Caffeine (g/100 g)	0.65 ± 0.06	1.03 ± 0.08	0.03 ± 0.01
Protein (g/100 g)	15.47 ± 0.78	20.87 ± 0.91	15.57 ± 0.79
Fat (g/100 g)	2.54 ± 0.18	1.70 ± 0.13	2.39 ± 0.17
Carbohydrates (g/100 g)	14.57 ± 0.28	9.45 ± 0.17	13.79 ± 0.38
Dietary fibre (g/100 g)	58.32 ± 5.02	57.18 ± 4.95	58.09 ± 5.00
Water (g/100 g)	2.00 ± 0.01	1.26 ± 0.02	1.84 ± 0.01

**Table 2 foods-11-00717-t002:** Formulation of the vegan biscuits. Quantities are reported in grams.

Ingredients	0CS	2CS	4CS	6CS
Wheat flour	420	411.6	403.2	394.8
Silverskin	0	8.4	16.8	25.2
Oat milk	120	120	120	120
Sucrose	140	140	140	140
Margarine	180	180	180	180
Baking powder	12	12	12	12
Salt	1	1	1	1

**Table 3 foods-11-00717-t003:** Values (means ± standard deviation) of moisture, water activity (a_w_), and CIELAB parameters (*L**, lightness; *h_ab_*, hue; *C*_ab_*, chroma) of the CS-added biscuits. Results of the analysis of variance (ANOVA) with Duncan’s post hoc test are reported both between different percentages of integration of CS (column) and the different types of silverskin (row).

	%CS	CSA	CSR	CSD	*Significance*
Moisture (%)	0	4.27 ± 0.18 ^a^	4.27 ± 0.18 ^a^	4.27 ± 0.18 ^a^	
	2	5.44 ± 0.12 ^bA^	7.19 ± 0.42 ^cbB^	6.69 ± 0.17 ^dB^	***
	4	5.61 ± 0.04 ^bA^	7.38 ± 0.15 ^cC^	6.34 ± 0.12 ^cB^	***
	6	6.05 ± 0.03 ^cB^	6.60 ± 0.06 ^bC^	5.77 ± 0.08 ^bA^	***
	*Significance*	***	***	***	
a_w_	0	0.31 ± 0.00 ^a^	0.31 ± 0.00 ^a^	0.31 ± 0.00 ^a^	
	2	0.41 ± 0.00 ^bA^	0.52 ± 0.00 ^cC^	0.47 ± 0.01 ^cB^	***
	4	0.42 ± 0.01^cA^	0.52 ± 0.00 ^cC^	0.46 ± 0.01 ^bcB^	***
	6	0.44 ± 0.05 ^dA^	0.49 ± 0.00 ^bB^	0.45 ± 0.01 ^bA^	***
	*Significance*	***	***	***	
*L**	0	56.78 ± 0.85 ^d^	56.78 ± 0.85 ^d^	56.78 ± 0.85 ^d^	
	2	53.57 ± 0.73 ^cC^	49.25 ± 0.89 ^cB^	43.17 ± 0.88 ^cA^	***
	4	50.26 ± 0.52 ^bC^	42.79 ± 0.75 ^bB^	35.02 ± 1.20 ^bA^	***
	6	47.43 ± 0.73 ^aC^	38.00 ± 0.56 ^aB^	30.67 ± 0.29 ^aA^	***
	*Significance*	***	***	***	
*h_ab_*	0	70.78 ± 0.44 ^a^	70.78 ± 0.44 ^b^	70.78 ± 0.44 ^d^	
	2	71.73 ± 0.51 ^bB^	72.77 ± 0.59 ^dC^	70.26 ± 0.36 ^cA^	***
	4	71.31 ± 0.29 ^bB^	71.95 ± 0.61^cC^	67.25 ± 0.27 ^bA^	***
	6	71.34 ± 0.39 ^bC^	70.17 ± 0.46 ^aB^	64.89 ± 0.42 ^aA^	***
	*Significance*	**	***	***	
*C*_ab_*	0	34.66 ± 0.45 ^d^	34.66 ± 0.45 ^d^	34.66 ± 0.45 ^d^	
	2	32.03 ± 0.60 ^cC^	29.77 ± 0.47 ^cB^	27.11 ± 0.36 ^cA^	***
	4	31.38 ± 0.17 ^bC^	27.86 ± 0.60^bB^	23.37 ± 0.59 ^bA^	***
	6	30.04 ± 0.40 ^aC^	26.74 ± 0.72 ^aB^	21.29 ± 0.24 ^aA^	***
	*Significance*	***	***	***	

CSA, Arabica coffee silverskin; CSR, Robusta coffee silverskin; CSD, decaffeinated coffee silverskin. Means followed by the same lower-case (columns) and upper-case (rows) letters are not significant different at *p* < 0.05. Significance: ** = *p* < 0.01; *** = *p* < 0.001; n.s. = not significant.

**Table 4 foods-11-00717-t004:** Results (means ± standard deviation) of weight loss, spread and hardness of the CS-added biscuits. Results of the analysis of variance (ANOVA) with Duncan’s post hoc test are reported both between different percentages of integration of CS (column) and between the different types of silverskin (row).

	%CS	CSA	CSR	CSD	*Significance*
Weight loss (g)	0	50.19 ± 3.54	50.19 ± 3.54	50.19 ± 3.54	
	2	53.62 ± 1.41	54.22 ± 1.42	51.40 ± 0.00	ns
	4	51.67 ± 1.04	54.60 ± 0.68	52.55 ± 0.78	ns
	6	48.46 ± 0.19 ^A^	52.34 ± 0.20 ^B^	54.05 ± 0.21 ^C^	***
	*Significance*	ns	ns	ns	
Spread	0	6.49 ± 0.21 ^b^	6.49 ± 0.21 ^b^	6.49 ± 0.21 ^c^	
	2	5.43 ± 0.18 ^aB^	4.15 ± 0.12 ^aA^	4.44 ± 0.18 ^aA^	**
	4	5.22 ± 0.03 ^a^	4.25 ± 0.43 ^a^	4.86 ± 0.02 ^b^	ns
	6	4.85 ± 0.30 ^a^	4.57 ± 0.18 ^a^	4.73 ± 0.04 ^a^	ns
	*Significance*	**	**	***	
Hardness (N)	0	64.92 ± 8.91 ^a^	64.92 ± 8.91	64.92 ± 8.91 ^a^	
	2	82.81 ± 18.88 ^aB^	75.65 ± 14.78	67.05 ± 5.02 ^a^	ns
	4	85.75 ± 10.69 ^bB^	70.36 ± 7.52 ^A^	104.76 ± 3.65 ^bC^	***
	6	67.12 ± 15.40 ^a^	65.85 ± 11.54	86.71 ± 28.15 ^ab^	ns
	*Significance*	*	ns	**	

CSA, Arabica coffee silverskin; CSR, Robusta coffee silverskin; CSD, decaffeinated coffee silverskin. Means followed by the same lower-case (columns) and upper-case (rows) letters are not significant different at *p* < 0.05. Significance: * = *p* < 0.05; ** = *p* < 0.01; *** = *p* < 0.001; ns = not significant.

**Table 5 foods-11-00717-t005:** Results of the Kruskal–Wallis test on consumer acceptance evaluation. Results are reported as sum of the ranks.

Attribute	Samples										*Sig.*
	0CS	2CSA	4CSA	6CSA	2CSR	4CSR	6CSR	2CSD	4CSD	6CSD	
Appearance	15,072 ^d^	14,904 ^cd^	11,880 ^abcd^	9184 ^ab^	10,448 ^ab^	10,032 ^ab^	8424 ^a^	13,024 ^bcd^	11,672 ^abcd^	10,800 ^abc^	***
Odour	14,328 ^cd^	11,080 ^abcd^	11,280 ^abcd^	10,920 ^abc^	10,056 ^abc^	9432 ^ab^	7840 ^a^	15,864 ^d^	13,480 ^bcd^	11,160 ^abcd^	***
Taste	14,696 ^b^	8424 ^a^	9240 ^a^	8792 ^a^	11,720 ^ab^	11,272 ^ab^	8952 ^a^	15,272 ^b^	14696 ^b^	12,376 ^ab^	***
Flavour	10,496 ^abc^	8920 ^ab^	8360 ^a^	10,864 ^abc^	10,176 ^ab^	11,888 ^abc^	11,112 ^abc^	14,904 ^c^	15,480 ^c^	13,240 ^bc^	***
Texture	13,032 ^bc^	9344 ^ab^	9040 ^ab^	8072 ^a^	10,776 ^abc^	10,256 ^ab^	11,376 ^abc^	14,776 ^bc^	15,480 ^d^	13,288 ^bc^	***
Overall Liking	14,576 ^bc^	9096 ^a^	8632 ^a^	9352 ^a^	10,960 ^ab^	11,472 ^abc^	9576 ^a^	15,304 ^c^	14,576 ^bc^	11,896 ^abc^	***
Purchase Interest	13,488 ^bcd^	8736 ^a^	8472 ^a^	9720 ^ab^	11,768 ^abcd^	11,896 ^abcd^	10,232 ^ab^	14,616 ^cd^	15,368 ^d^	11,144 ^abc^	***

CSA, Arabica coffee silverskin; CSR, Robusta coffee silverskin; CSD, decaffeinated coffee silverskin. Means followed by the same lower-case letters are not significant different at *p* < 0.05. Significance: *** = *p* < 0.001.

**Table 6 foods-11-00717-t006:** Values (means ± standard deviation) of total phenolic content (TPC) and radical scavenging activity (RSA) of the CS-added biscuits. Results of analysis of variance (ANOVA) with Duncan’s post hoc test are reported both between different percentages of integration of CS (column) and the different types of CS (row).

	% CS	CSA	CSR	CSD	*Significance*
TPC (mg GAE/g)	0	0.41 ± 0.02 ^a^	0.41 ± 0.02 ^a^	0.41 ± 0.02 ^a^	
	2	0.42 ± 0.01 ^aB^	0.41 ± 0.00 ^aA^	0.72 ± 0.00 ^bC^	***
	4	0.47 ± 0.01 ^bA^	0.44 ± 0.01 ^bA^	1.03 ± 0.03 ^cB^	***
	6	0.49 ± 0.01 ^bA^	0.57 ± 0.01 ^cB^	1.36 ± 0.01 ^dC^	***
	*Significance*	***	***	***	
RSA (μmol TE/g)	0	0.43 ± 0.11 ^a^	0.43 ± 0.11 ^b^	0.43 ± 0.11 ^a^	
	2	0.93 ± 0.06 ^bB^	0.21 ± 0.01 ^aA^	1.13 ± 0.01 ^bC^	***
	4	1.42 ± 0.05 ^cB^	0.34 ± 0.02 ^bA^	2.60 ± 0.01 ^cC^	***
	6	1.82 ± 0.06 ^dB^	0.71 ± 0.04 ^cA^	3.40 ± 0.03 ^dC^	***
	*Significance*	***	***	***	

GAE, gallic acid equivalents; TE, Trolox equivalents. Means followed by the same lower-case (columns) and upper-case (rows) letters are not significant different at *p* < 0.05. Significance: *** = *p* < 0.001.

## Data Availability

Data is contained within the article or Appendix A.

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
