# Peer review of "Coffee Silverskin as a Functional Ingredient in Vegan Biscuits: Physicochemical and Sensory Properties and In Vitro Bioaccessibility of Bioactive Compounds"

_foods, 2022, doi:10.3390/foods11050717_

Round 1

Reviewer 1 Report

The manuscript proposed the use of coffee silverskin as a functional ingredient in vegan biscuit. The topic is of interest .  Some comments and suggestions are given below:

Lines 103-105: Remove sentence or provide reference for this information.  To my knowledge he most used method industrially is still extraction with steam

Line 134 – provide information on the decafeination method used by Lavazza

Line 165 - since the hedonic scales are different, the specific levels should be presented in the supplementary material

Line 211 - how was the concentration measured? was it based on TPC? if so, this methodology must be presente before item 2.7 and the authors should comment on the limitation of this method, because it has many interferences and not only phenolics are measured (this was only mentioned at the end of the manuscript)

Lines 240-242 – merge paragraphs

Line 244 – Regarding moisture…

Lines 258-261 - remove  or modify sentence… product is probably safe from a microbiological point of view, BUT no microbiological tests were performed,

Line 285 - Discussion should enphasize the significant variation in luminosity with adding CS. This is clearly seen in Figure 1

Lines 286-299 - Discussion needs to be rewritten. It is common in the literature to refer to a* and b* in terms of yellowness and redness, but this analysis is not appropriate because one parameter affects the other. For example, a positive b* value can be only associated to yellowness if a* tends to zero, and vice-versa for a* related to redness. I recommend calculation of the parameters chroma (color saturation) and hue (color tone), corresponding to radius and angle in polar coordinates considering the a* b* axis. This way the discussion can be based on actual variations in color (see attached file)

Line 300 – With regard to the DE parameter…

Lines 327-329 - remove this discussion... if there is no statistical difference these percentages can not be taken into account

Line 333 – replace attitude by behavior

Line 342 - replace latter by fiber

Line 392 - This discussion is superficial and needs to be improved. Results must also be provided in terms of the hedonic scale (average score values), so we can see the actual effect of adding CS on the sensory parameters. Also, check the score of 6CSA (Table 5) in terms of purchase interest... it appears there was a mistype.

Lines 449-451 - this comparison can not be made, please remove

Line 476 – may be possibly credited to

Line 586 - I believe that a sentence comenting on the need to evaluate the phenolic profile should be interesting here

Author Response

We would like to thank the Reviewer for appreciating and greatly improving our work with their commentary. A list explaining how we addressed each comment/suggestion has been prepared.

Reviewer 2 Report

The authors describe the characterization of different properties of biscuits enriched with coffee silverskin. The study is well described and the conclusions are backed up by the data.

The following remarks might be considered:

Line 15: levels

Line 18: „healthy“ is somehow an unscientific word, especially considering that no clinical studies were conducted

Line 27: drinks

Line 29: check correct botanical name for canephora. Should not be “L.” but “Pierre ex Froehner”.

Line 45: consider adding reference on composition of silverskin: Gottstein, V.; Bernhardt, M.; Dilger, E.; Keller, J.; Breitling-Utzmann, C.M.; Schwarz, S.; Kuballa, T.; Lachenmeier, D.W.; Bunzel, M. Coffee Silver Skin: Chemical Characterization with Special Consideration of Dietary Fiber and Heat-Induced Contaminants. Foods 202110, 1705. https://doi.org/10.3390/foods10081705

Line 87: I believe that the health effects are a bit exaggerated

Line 134: can the proportions be specified?

Line 140: please provide details on the composite ingredients, i.e. please specify the types and ingredients of margarine, oar milk, and wheat flour.

Line 498: consider adding the supplementary material as “Annex” in the main file (check journal template)

Line 514 and throughout: please specify only significant decimals

Line 586, 588 and throughout: consistently use in vitro and in vivo (no italics, no dash)

Author Response

(The authors gave the same response as above.)
